# Impact of Gas Diffusion Layer Compression on Electrochemical Performance in Proton Exchange Membrane Fuel Cells: A Three-Dimensional Lattice Boltzmann Pore-Scale Analysis

**DOI:** 10.3390/nano14242012

**Published:** 2024-12-14

**Authors:** Hao Wang, Xiaoxing Yang, Guogang Yang, Guoling Zhang, Zheng Li, Lingquan Li, Naibao Huang

**Affiliations:** 1Marine Engineering College, Dalian Maritime University, Dalian 116026, China; whdlmu@dlmu.edu.cn (H.W.); yangxiaoxing@dlmu.edu.cn (X.Y.); li_0203@dlmu.edu.cn (L.L.); nbhuang@dlmu.edu.cn (N.H.); 2School of Marine Engineering, Guangzhou Maritime University, Guangzhou 510725, China; zhangguoling@gzmtu.edu.cn; 3Guangdong Provincial Key Laboratory of Intelligent Equipment for South China Sea Marine Ranching, Guangdong Ocean University, Zhanjiang 524088, China; li.zheng@gdou.edu.cn

**Keywords:** PEMFC, gas diffusion layer, compression, lattice Boltzmann method, electrochemical performance

## Abstract

Proton exchange membrane fuel cells (PEMFCs) are being pursued for applications in the maritime industry to meet stringent ship emissions regulations. Further basic research is needed to improve the performance of PEMFCs in marine environments. Assembly stress compresses the gas diffusion layer (GDL) beneath the ribs, significantly altering its pore structure and internal transport properties. Accurate evaluation of the PEMFC cathode’s electrochemical performance at the pore scale is critical. This study employs a three-dimensional multicomponent gas transport and electrochemical reaction lattice Boltzmann model to explore the complex interplay between GDL compression and factors such as overpotential, pressure differential, porosity, and porosity gradient on PEMFC performance. The findings indicate that compression accentuates the reduction in oxygen concentration along the flow path and diminishes the minimum current density. Furthermore, compression exacerbates the reduction in current density under varying pressure conditions. Increased local porosity near the catalyst layer (CL) enhances oxygen accessibility and water vapor exclusion, thereby elevating the mean current density. Sensitivity analysis reveals a hierarchy of impact on mean current density, ranked from most to least significant: overpotential, porosity, compression, porosity gradient, and pressure difference. These insights into the multicomponent gas transfer dynamics within compressed GDLs inform strategic structural design enhancements for optimized performance.

## 1. Introduction

Hydrogen energy, recognized as the ultimate clean and efficient renewable energy source, plays a crucial role in building a green and low-carbon industrial system. Its development promotes environmental sustainability and fosters a resource-saving, eco-friendly society. Currently, fuel cells serve as both primary and backup power sources for commercial, industrial, and residential buildings and remote areas [1,2]. They also power various vehicles, including forklifts [3], automobiles [4], ships [5], motorbikes [6], and submarines [7]. Proton exchange membrane fuel cells (PEMFCs) are particularly favored for their fast start-up, low operating temperature, high technological maturity, and advanced commercialization. However, the widespread adoption and commercialization of PEMFC technology still face several challenges, necessitating further optimization and improvement in power density, durability, and manufacturing costs [1].

To achieve a suitable voltage output, the individual cells of a PEMFC must be assembled into a stack and mechanically connected to ensure effective sealing and current collection [8]. The carbon paper gas diffusion layer (GDL), with its typical porous structure, tends to deform under compression. Due to the geometric shape of the bipolar plate, the carbon fibers beneath the ribs are squeezed, reducing the GDL’s porosity in these regions as the clamping force increases. Conversely, the area below the gas channel experiences much less compressive force, leaving the GDL structure in this region largely unaffected. The disparity in compression forces across different regions can lead to shearing and fragmentation of the GDL at the rib–channel interface under high compression, resulting in more severe changes to the pore structure [9]. These compression-induced changes in microstructure and pore size distribution significantly impact the gas transfer characteristics within the GDL.

Currently, the finite element method (FEM) or finite volume method is primarily used for studying the transport processes in PEMFCs at the macroscopic scale. However, these methods often neglect internal structural details, considering only physical characteristic parameters such as porosity and catalytic reaction rates. Chippar et al. [10] investigated GDL compression in a 3D two-phase PEMFC model and observed that reduced GDL porosity and permeability due to compression led to increased liquid water saturation within the GDL, causing a localized decrease in current density under the ribs and exacerbating the non-uniformity of current density distribution. Hottinen et al. [11] employed FEM to study the impact of non-uniform GDL compression on temperature distribution in PEMFCs. The study revealed that heat generated in electrodes under channels was laterally transferred into the GDL region under ribs with higher thermal conductivity, resulting in significant temperature gradients within electrodes. However, macro-scale modeling has not adequately accounted for the impact of compression on GDL gas transport parameters, including permeability and diffusivity. Yan [12] and Hottinen [11] estimated the permeability of compressed GDL using the Kozeny–Carman relationship. The diffusivity of GDL after compression was predicted by porosity, tortuosity, and pore size in the work of Wang et al. [13]. The tortuosity was predicted using an empirical relation proposed by Barrande et al. [14]. However, this relation is mainly for porous particles and cannot accurately predict the tortuosity of carbon paper GDL. Hottinen et al. [15] predicted the permeability of compressed GDL using a cubic fit based on GDL thickness. In realistic situations, the transport characteristics due to the mixing and influence of components in PEMFCs are also crucial. Therefore, to more effectively understand the mechanisms and characteristics of fluid flow and transfer in compressed GDLs, it is necessary to study them from the microscopic pore scale, considering the complex pores inside the compressed GDL.

To study the pore-scale transfer characteristics within a GDL, it is first necessary to obtain its microstructure. Microstructural modeling can reveal the microstructural characteristics of the GDL [16]. X-ray computed tomography (XCT) and stochastic reconstruction methods are commonly used to construct the 3D microstructure of GDL [17]. XCT generates a tomographic image of a material from computer-processed X-rays, enabling the internal structure of a material to be obtained without destruction, creating a series of two-dimensional tomographic images that combine to form a 3D image of the material [18]. A better understanding of the mass transfer characteristics of GDL can be achieved by establishing pore structures with different pore sizes and shapes. However, experimental techniques are often costly and time-consuming. The stochastic reconstruction model uses a stochastic generator to reconstruct a virtual porous structure representing a real GDL material based on structural information from the carbon paper manufacturing process. A general stochastic reconstruction method for carbon paper GDL was first proposed by Schulz et al. [19], which assumes that the fibers can overlap in a plane perpendicular to the main flow direction and are reduced to straight cylinders. This method has been widely used for pore-scale studies of GDL due to its high efficiency, speed, low cost, and ease of modifying structural parameters [20,21].

Despite the potential of PEMFC technology, gas supply difficulties within the cathode hinder wider commercialization [22]. Air enters the PEMFC from the gas channel inlet, distributing across the GDL surface through bipolar plate channels, and flows into the GDL and catalyst layer (CL) driven by pressure gradients, necessitating efficient gas transfer processes for improved cell performance. Innovative flow field designs, like Nguyen’s [23] interdigitated flow channels, enhance gas transfer efficiency by forcing gas through the GDL via forced convection instead of diffusion, improving gas supply to the CL [24]. The anisotropic GDL transfer characteristics are critical when using such designs, as gas flows downward into the GDL under inlet/outlet pressure differences, enhancing transverse flow effects below rib plates. The lattice Boltzmann method (LBM) has advanced sufficiently for pore-scale PEMFC studies, including transport parameter calculations [25,26], single-phase multicomponent gas transfer [27,28], liquid water transport [29,30], and electrochemical reactions [31,32]. Molaeimanesh and Akbari [33] developed a 3D pore-scale LBM model for the PEMFC cathode, capable of simulating anisotropic GDL gas flow and CL electrochemical reactions. Bahoosh et al. [34] used this model to study GDL structural parameters’ effects on water vapor, oxygen, and current density distributions. Molaeimanesh et al. [35] further analyzed activation overpotential, inlet/outlet pressure differences, and rib-to-channel width ratios, highlighting GDL microstructure’s significant impact on oxygen and current density distributions, and reconstructed three carbon cloth GDLs with varied microstructures to simulate reactive airflow through carbon cloth, finding increased difficulty for oxygen penetration into the CL with denser fiber bundles, reducing current density.

Due to clamping forces applied during assembly, GDL deformation at the pore scale alters its anisotropic transfer characteristics. Understanding these changes in relation to compression is crucial for optimizing PEMFC design. However, the single-phase transfer characteristics within compressed GDLs are not well understood. This paper develops a 3D single-phase multicomponent gas transport LBM model to explore the effects of overpotential, inlet/outlet pressure differences, porosity, and porosity gradients on gas flow in compressed GDLs, evaluating their influence on electrochemical performance.

## 2. Methodology

### 2.1. Stochastic Reconstruction Modeling of Compressed GDL

In this section, the carbon paper GDL is reconstructed, and the reconstruction of GDL is based on the idea of using carbon fibers with specified diameters to form a single layer by randomly combining them in the same plane interval and then superimposing a series of carbon fiber layers to obtain the GDL of the target structural parameters. It is necessary to selectively carry out some reasonable simplifications; for example, the current common simplification measures include (1) assuming that the carbon fibers are long straight cylinders, which are combined in the same plane to form a single layer, and the carbon fibers interspersed along the direction of the thickness are not taken into account; (2) assuming that the diameters of the carbon fibers are the same and constant; and (3) assuming that the carbon fibers in the same layer are allowed to overlap with each other. The GDL can be considered as a superposition of multiple layers of cylindrical carbon fibers, and each cylinder can be considered as a collection of points whose distance from the axis is less than the radius. Each axis is defined by two points at arbitrary positions on different boundaries in the plane. Firstly, the structural parameters of the reconfigured carbon fiber skeleton need to be determined, such as the fiber diameter *d*, the edge length *L*, and the number of fiber layers *k*. Some other parameters can be calculated using the following equations [36]:(1)s=kd
(2)$$m(n)=\left(1-\varepsilon_{loc,tar}(n)\right)\frac{4L}{\pi d}$$
(3)$$\varepsilon =1-\frac{N}{M}$$
where *s* is the thickness of the carbon fiber skeleton, *N* is the number of matrix points in the cylinder, and *M* is the total number of matrix points in the computational domain. The *z*-coordinate of the center plane of each carbon fiber layer is first calculated, the axis is determined from two random points, and then the cylinder is generated based on the diameter of the carbon fibers and the axis. The process is repeated *m* times to obtain *m* fibers, and then the porosity of the fiber layer *ε_local_* is calculated. When the deviation between each *ε_local_* and the target porosity of the fiber layer is less than 0.01, the next layer of carbon fibers is generated. When all *k* layers of carbon fibers are generated, the total porosity *ε_total_* is calculated, and when the deviation from *ε_total,tar_* is less than 0.01, the GDL satisfying the requirements is obtained.

Froning et al. [21] proposed a compression method by merging the adjacent fiber layers with the largest porosity, but it results in a loss of the solid phase. We optimized the algorithm to achieve a maximum error of less than 0.5% between the actual porosity of the compressed GDL and the theoretical value when the GDL is compressed by 40%. The compression ratio (CR) is defined as the ratio of the reduced thickness to the initial thickness. Details of the optimized compression algorithm can be seen in our previous work [27].

### 2.2. Single-Phase Multicomponent Lb Model

The LBM has the advantage of dealing with complex boundary conditions and can directly simulate the flow field in a connected porous medium with complex geometrical boundaries. The LBM does not need to solve the partial differential equations, and the fluid flow parameters are calculated from the equations of the state of the nodes of the flow field. The discrete velocity evolution equation can be expressed as follows:(4)$$f_{i}(x+c_{i}\Delta t,t+\Delta t)=f_{i}(x,t)-\omega [f_{i}(x,t)-f_{i}^{eq}(x,t)]$$
(5)$$f_{i}^{eq}(x,t)=w_{i}\rho \left[1+\frac{c_{i}u}{{c_{s}}^{2}}+\frac{{\left(c_{i}u\right)}^{2}}{2{c_{s}}^{2}}-\frac{u^{2}}{2{c_{s}}^{2}}\right]$$
where *f_i_* is the velocity distribution function and $f_{i}^{eq}$ is the velocity equilibrium distribution function. In this paper, the D3Q19 model is used to simulate single-phase multicomponent gas flow in GDL, and the particles at each node have 19 possible directions of motion at the next moment. *ω* is the relaxation frequency, *c_i_* is the lattice velocity direction, and *w_i_* is the weighting factor of each velocity direction. *ρ* denotes the fluid density and *u* denotes the fluid velocity.
(6)$$c_{i}=\left\{\begin{array}{l} (0,0,0),i=0\\ (\pm 1,0,0),(0,\pm 1,0),(0,0,\pm 1),i=1–6\\ (\pm 1,\pm 1,0),(0,\pm 1,\pm 1),(\pm 1,0,\pm 1),i=7–18 \end{array}\right.$$
(7)$$w_{i}=\left\{\begin{array}{l} \frac{1}{9},i=0\\ \frac{1}{18},i=1–6\\ \frac{1}{36},i=7–18 \end{array}\right.$$
(8)$$\rho =\sum_{i=0}^{18}f_{i}$$
(9)$$u=\frac{\sum_{i=0}^{18}f_{i}c_{i}}{\rho }$$

When dealing with boundary conditions, it is often necessary to compute unknown distribution functions on the boundary from macroscopic physical quantities on the boundary. The schematic of the computational domain is shown in Figure 1. The *x* = 0 and *x* = *L* planes are the inlet of dry air and the outlet of the mixture, respectively, setting the pressure boundaries proposed by Zou and He [37]. The periodic boundary conditions are applied to the *y* = 0 and *y* = L planes. The top of the computational domain is the lower surface of the rib plate, and the stress in the -*z* direction is applied to the GDL. A half-step bounce-back boundary condition is applied to the carbon fiber surface. To enable the simulation of oxygen being consumed in the CL, the CL is reduced to a thin boundary at the bottom of the computational domain, and the modified bounce-back boundary condition proposed by Kamali et al. is applied. As particles of component A hit the reaction surface, the fraction of them that are $K_{sr}^{LB}$ are converted to component B, while the remaining portion of the (1 − $K_{sr}^{LB}$) component A particles do not participate in the electrochemical reaction and bounce back into the flow domain in the form of a bounce-back boundary condition. Here, $K_{sr}^{LB}$ is the dimensionless surface reaction constant. The reaction rate at the reaction wall is as follows [38]:
(10)$$r=K_{sr}^{LB}\phi$$

After derivation, the surface reaction constant of the lattice unit is obtained as follows:(11)$$K_{sr}^{LB}=\left(\frac{6K_{sr}\Delta t}{\Delta x}\right)/\left(1+\frac{K_{sr}\Delta x}{2D}\right)$$
where *D* is the diffusion coefficient and *K_sr_* is the surface reaction rate constant, determined from the current density (*J*) and the oxygen concentration at the surface of the cathode CL (*ρ_O_*_2_):(12)$$K_{sr}=\frac{J}{4F\rho_{O2}}$$
where *F* is Faraday’s constant, and the current density can be obtained from the Butler–Volmer equation:(13)$$J=a_{r}J_{ref}\frac{\rho_{O2}}{\rho_{O,ref}}\left[\exp\left(\frac{\alpha_{f}F\eta }{R_{u}T}\right)-\exp\left(-\frac{\alpha_{r}F\eta }{R_{u}T}\right)\right]$$

For simulating the oxygen reduction reaction, it is required to apply the modified bounce-back boundary conditions on the CL surface, and the distribution function of the unknowns on the boundary can be obtained from the following equation, which is available for oxygen:(14)$$f_{O_{2},i}=(1-K_{sr}^{LB})f_{O_{2},\bar{i}}$$

And for water vapor, the distribution function is below:(15)$$f_{H_{2}O,i}=\frac{2M_{H_{2}O}}{M_{O_{2}}}K_{sr}^{LB}f_{O_{2},\bar{i}}+f_{H_{2}O,\bar{i}}$$
where *i* denotes the ordinal number of the velocity direction away from the reaction boundary, and $\bar{i}$ denotes the opposite velocity direction to *i*.

The model is based on the following assumptions: (1) dry air flows in from the inlet, and the partial pressure of water vapor under dry conditions is low enough to make it difficult to reach the local water saturation pressure and condense, so water is assumed to be in the gas phase; (2) the operating conditions are isothermal and steady-state; and (3) it is assumed that the gas mixture described is the ideal gas mixture. More specific details on model validation are given in our previous work.

## 3. Results and Discussion

Compression changes the microstructure of the GDL and the mass transport process within it, and the evaluation of the electrochemical performance of the PEMFC cathode at the pore scale has been understudied in the past, and, in particular, the effects of compression and the structural parameters of the GDL, as well as the operating conditions of the fuel cell, on the electrochemical performance of the cathode have not yet been investigated. In this work, we mainly investigated the effects of different porosities, gradient distributions of porosity, overpotentials, pressure difference coupled compression effects on the electrochemical performance of the cathode, and mass transport within the cathode GDL, simulated the flow of reactive gases through the GDL to the surface of CL for the reaction, and analyzed the distribution of the concentration of oxygen and water vapor within the GDL as well as the distribution of the current density at the surface of CL. The base and calculated cases for the considered parameters are listed in Table 1, and compression ratios of 0, 10%, 20%, 30%, and 40% are considered for each case. *ε*_n=1_ denotes the local porosity of the first carbon fiber layer of the GDL near the CL side.

### 3.1. Effect of Overpotential and Compression Ratio on Electrochemical Performance

Overpotential is one of the irreversible factors of the reduction reaction, and the distribution of oxygen concentration within the GDL for different overpotentials and compression ratios is shown in Figure 2a. The concentration of oxygen decreases gradually from the inlet to the *x* = L plane as oxygen is consumed on the CL. According to Equation (13), as overpotential increases, the electrochemical reaction rate increases and the oxygen consumption increases, so the oxygen concentration in the GDL gradually decreases. Compression makes the decrease in oxygen concentration along the flow direction more pronounced, and for different overpotential arithmetic cases, when the compression ratio is increased, a significant decrease in the oxygen concentration at the outlet can be observed. The distribution of water vapor concentration in the GDL under different overpotentials and compression ratios is presented in Figure 2b. Water vapor is generated in the CL, so the concentration of water vapor is higher near the surface of the CL, and the water vapor is then wrapped by the gas mixture and collects at the outlet, so the inlet region has a low concentration and the outlet region has a high concentration in the *x*-direction. The pore space of GDL is reduced after being compressed, and it is more difficult for the water vapor generated to be carried away; with the increase in the compression ratio, the concentration of water vapor in the *x* = L plane increases significantly.

As can be seen in Figure 3a, the current density increases with increasing overpotential. For the uncompressed samples, as *η* = 0.3 V, 0.35 V, 0.4 V, 0.45 V, and 0.5 V, the current densities are distributed at 0.3782–0.3863 A/cm^2^, 0.8518–0.8784 A/cm^2^, 1.8954–1.9960 A/cm^2^, 4.1134–4.5294 A/cm^2^, and 8.4920–10.2484 A/cm^2^; the difference between the maximum and minimum current densities increases as the overpotential increases. For different overpotentials, the minimum current density decreases by 2.10%, 3.04%, 5.04%, 9.18%, and 17.14%, respectively, compared to the maximum current density, indicating that an increase in the activation overpotential also leads to an increase in the inhomogeneity of the current density distribution. Increasing compression leads to a smaller minimum current density and a more inhomogeneous current density distribution. Taking the base case as an example, when the compression ratios are 10%, 20%, 30%, and 40%, the minimum current densities decrease by 0.33%, 0.95%, 1.69%, and 3.13%, respectively, compared to the uncompressed samples. That is to say, when the GDL is locally compressed more, the performance is worse.

The mean current density on the CL surface for different overpotentials and compression ratios is shown in Figure 3b. The effect of overpotential on the current density is evident as the mean current density increases with increasing overpotential at any compression level. An increase in the compression ratio resulted in a decrease in the mean current density, and the decrease in the mean current density due to the compression of the GDL is not significant at low overpotentials, whereas the compression effect becomes sharp at *η* = 0.5 V. Especially at high compression ratios, such as at 40% compression, the mean current densities decreased by 0.53%, 1.07%, 2.15%, 4.28%, and 8.04%, respectively, compared to the uncompressed condition, which indicates that the operation of the fuel cell at high overpotentials requires the compression ratio to be controlled in a reasonable range, and that high compression ratios lead to higher performance degradation.

### 3.2. Effect of Pressure Difference and Compression Ratio on Electrochemical Performance

The distribution of oxygen concentration inside the GDL under different pressure difference and compression ratio conditions is shown in Figure 4a. Similarly, as oxygen is consumed at the CL, the closer to the CL, the lower the oxygen concentration in the direction perpendicular to the CL. For different pressure differences, since the inlet pressure is kept constant at 1.5 atm, the higher the pressure difference, the lower the gas outlet pressure. However, for the 25 samples with different inlet and outlet pressure differences, the oxygen concentration in the GDL ranges from 9.8113 to 10.9350 mol/m^3^, which implies that the supply of oxygen is sufficient, and no oxygen deficiency is observed for the several pressure difference conditions considered in this section. As the outlet pressure decreases due to an increase in pressure difference, the outlet oxygen partial pressure also decreases, so for the uncompressed sample, the oxygen concentration in the GDL decreases with an increase in pressure difference. The distribution of water vapor concentration within the GDL at different pressure differences and compression ratios is shown in Figure 4b. The change in water vapor concentration due to pressure difference is not significant, and the distribution of water vapor concentration is closer at any compression ratio. The effect of compression on the distribution of water vapor concentration, however, is more intuitive, with higher compression leading to higher water vapor concentrations near the CL surface as the compression ratio increases, due to the presence of a more compact microstructural barrier preventing the diffusion of water vapor away from the GDL when compressed.

Figure 5a illustrates the current density distribution on the CL surface for different pressure differences and compression ratios. The distribution arrangement of the carbon fibers has an impact on the flow behavior of the reactive gas and, consequently, on the distribution of the current density. The minimum value of current density occurs at the upper right, i.e., at the gas outlet, *y* = 0.75 L, for all conditions, which is due to the crossing of carbon fiber at this location, and the intersection of multiple carbon fibers makes the oxygen supply in this region not satisfactory, with a consequent decrease in current density. For the uncompressed samples, the maximum value of current density is 1.9960 A/cm^2^ for Δ*P* = 0.0025 atm, 0.005 atm, 0.01 atm, 0.02 atm, and 0.04 atm, while the minimum values of current density are 1.9070 A/cm^2^, 1.9030 A/cm^2^, 1.8954 A/cm^2^, 1.8811 A/cm^2^, and 1.8549 A/cm^2^, respectively. The current density decreases with increasing pressure difference, and increasing pressure difference leads to stronger current density inhomogeneity. Increasing compression leads to a smaller minimum current density and a more inhomogeneous current density distribution, and as the compression ratio increases, the area of low current density represented by the blue in the outlet region is more pronounced.

The mean current density on the CL surface for different pressure difference and compression ratio conditions is illustrated in Figure 5b. The mean current density decreases with the increase in the pressure difference, and the increase in the compression ratio makes the mean current density decrease, and the compression leads to a closer decrease in the current density values for different pressure difference conditions. For example, at 40% compression, the mean current density decreased by 0.0413 A/cm^2^, 0.0415 A/cm^2^, 0.0418 A/cm^2^, 0.0429 A/cm^2^, and 0.0456 A/cm^2^, respectively, compared to the uncompressed condition. At compression ratios of 10%, 20%, and 30%, the mean current density decreases in an approximately linear trend, while a sudden drop in current density can be observed when the compression ratio is increased to 40%, indicating that the microstructural barrier resulting from high compression ratios further weakens the oxygen flow and diffusion ability, and the fuel cell performance decreases significantly when the cathode operates at high compression ratio conditions.

### 3.3. Effect of Porosity and Compression Ratio on Electrochemical Performance

Figure 6a shows the current density distribution on the CL surface under different porosity and compression ratio conditions. It can be clearly observed at compression ratios of 20% and above that the current density at the location where the carbon fiber is located is lower than that at the void, which is due to the fact that the presence of the carbon fibers will have a hindering effect on the oxygen transport, and the current density distribution is not uniform at the location where the carbon fibers are located and at its two sides. As the porosity increases, it implies a larger pore size distribution and less resistance to oxygen transport, generating higher current densities. The GDLs with low porosity conditions are more sensitive to compression, and for *ε* = 0.66 and 0.72 GDLs, the decrease in current density at the carbon fiber crossings is more drastic with the increase in compression ratio, especially when *ε* = 0.66 GDLs are compressed by 40%, which results in the formation of a distinctive low-current-density zone in the outlet region. In contrast, for the high porosity GDL, even when compressed, the continuous oxygen supply resulted in no significant decrease in current density.

The mean current density of the CL surface under different porosity and compression ratio conditions is demonstrated in Figure 6b. For uncompressed GDL, the mean current densities are 1.9078 A/cm^2^, 1.9304 A/cm^2^, 1.9375 A/cm^2^, 1.9426 A/cm^2^, and 1.9526 A/cm^2^ for *ε* = 0.66, 0.72, 0.78, 0.84, and 0.90, respectively, and the mean current density increases with the increase in porosity. For GDLs of all porosities, compression leads to a decrease in mean current density; however, the sensitivity of mean current density to compression is not the same for GDLs of different porosities, and GDLs with smaller porosities produce a greater current density decay when subjected to compression. For GDLs with *ε* = 0.66, the current density decreases by 0.49%, 1.39%, 2.88%, and 5.51% when subjected to compression by 10%, 20%, 30%, and 40%, respectively, while for GDLs with *ε* = 0.90, they decrease by 0.15%, 0.25%, 0.43%, and 0.63%, respectively. Therefore, GDLs need to be designed with more attention to the uniformity of porosity distribution, and GDLs with lower local porosity are more likely to have oxygen deficiency and liquid water generation after being compressed, which will also result in lower current density and decreased cathode performance.

### 3.4. Effect of Porosity Gradient and Compression Ratio on Electrochemical Performance

Designing porosity gradient GDLs has become one of the important research focuses regarding GDLs. A suitable porosity gradient has improved the permeability, effective diffusivity, and effective thermal conductivity of GDLs. To further investigate the effect of porosity gradient distribution on cathode mass transport in fuel cells, four porosity gradient GDLs of step type with *ε*_n=1_ = 0.66, 0.72, 0.84, and 0.90 are generated. The current density distribution on the CL surface under different porosity gradients and compression ratios is given in Figure 7a. For uncompressed GDLs, the current densities of the samples with *ε*_n=1_ = 0.66, 0.72, 0.78, 0.84 and 0.90 are in the 1.8757–1.9960 A/cm^2^, 1.8802–1.9960 A/cm^2^, 1.8954–1.9960 A/cm^2^, 1.9167–1.9960 A/cm^2^ and 1.9304–1.9960 A/cm^2^ ranges. The maximum current density occurs at the inlet and has the same maximum current density because the oxygen inlet concentration is the same across the samples; the smaller local porosity of the GDL on the CL side results in a smaller minimum current density and a more heterogeneous distribution of current densities. As the compression ratio increases, the current density tends to decrease and in a more heterogeneous manner, with the minimum current densities resulting from the five porosity gradient GDLs being 1.8284 A/cm^2^, 1.8301 A/cm^2^, 1.8361 A/cm^2^, 1.8622 A/cm^2^, and 1.8894 A/cm^2^ when being compressed by 40%, for example.

The mean current densities of the CL surface under different porosity gradients and compression ratios are given in Figure 7b. The trend of the mean current densities with porosity gradients and compression ratios is in agreement with that shown in Figure 7a. The larger local porosity on the CL side facilitates oxygen replenishment and water vapor exclusion and hence has a higher mean current density. The mean current densities of the samples with *ε*_n=1_ = 0.66, 0.72, 0.78, 0.84, and 0.90 are 1.9231 A/cm^2^, 1.9308 A/cm^2^, 1.9375 A/cm^2^, 1.9507 A/cm^2^, and 1.9580 A/cm^2^ when they are not compressed, respectively. Compression also leads to a decrease in the mean current density, which decreases with the compression ratio in a variable trend due to the non-uniformity of the porosity distribution in the porosity gradient GDL. For GDLs with *ε*_n=1_ = 0.66 and 0.72, the mean current densities are significantly smaller than those of the uniform GDLs at compression ratios of 0, 10%, and 20%, while at high compression ratios, 30% and 40%, the mean current densities are very close to those of the uniform GDLs. This is due to the fact that under compression, the carbon fiber layer with larger local porosity is less strong and preferentially displaced, which makes the porosity distribution of the compressed GDL gradually homogeneous, and thus exhibits cathodic properties close to those of the uniform GDL at high compression ratios.

### 3.5. Parameter Sensitivity Analysis

The effects of various parameters on the mass transfer process within the GDL have been described and analyzed in the previous section, and to facilitate the comparison of the effects of different parameters on the electrochemical performance of the cathode of the fuel cell, the degree of influence of each parameter on the mean current density at the surface of the CL has been calculated based on the results obtained. The sensitivity of each parameter to the mean current density on the CL surface is evaluated by calculating the influence degree ζ, which is defined as follows [29]:(16)$$\zeta =\sum \frac{\sum_{1}^{n}\frac{\left|J-J_{default}\right|}{J_{default}}}{n}$$
where *J_default_* indicates the mean current density corresponding to the base condition, J indicates the mean current density when each parameter is varied, and *n* is the number of samples corresponding to the condition.

Figure 8 shows the degree of influence of each parameter on the mean current density. Based on the calculations described earlier, the sensitivity to the mean current density is in the order of overpotential > porosity > compression > porosity gradient > pressure difference. Among them, the degree of influence of overpotential is much higher than that of other parameters, so it is effective to keep the fuel cell operating at a suitable overpotential to enhance the fuel cell performance. The degree of influence of pressure difference is the smallest among all the parameters, indicating that the variation in pressure has less effect on the mean current density. The effect of compression on the mean current density is slightly less than porosity, and the mean current density decreases with an increasing compression ratio. The sensitivity analysis of different parameters on the mean current density in this section can provide a reference for predicting the cathode mass transport capacity and electrochemical performance of GDL during the design process.

## 4. Conclusions

This study systematically developed a three-dimensional lattice Boltzmann model for multicomponent transport and electrochemical reactions to scrutinize the mass transfer characteristics within the compressed gas diffusion layer (GDL) of proton exchange membrane fuel cells. We focused on the dynamics of multicomponent gas transfer and electrochemical reactions within the GDL, taking into account variables such as overpotentials, differential pressures, porosities, and porosity gradients under compression. The key findings include the following:(1)Both overpotentials and compression ratios critically influence fuel cell performance. Higher overpotentials lead to significant alterations in oxygen consumption and water vapor generation, resulting in more uneven distributions of current density. These changes are exacerbated by increased compression ratios, which further degrade performance. Notably, the combination of high overpotentials and elevated compression ratios causes especially severe performance declines.(2)Both factors markedly affect the gas flow within the GDL. Rising pressure differences reduce the oxygen concentration at the outlet, while greater compression ratios not only decrease outlet oxygen concentration but also enhance water vapor concentration. A compression ratio of 40% results in a sharp decline in mean current density, suggesting that the microstructural barriers introduced by high compression impair oxygen flow and diffusion, thereby significantly diminishing fuel cell performance.(3)At compression ratios of 20% or higher, the current density at locations containing carbon fibers is substantially lower than that at voids. The mean current density on the catalyst layer (CL) surface increases with porosity but undergoes more significant decay under compression.(4)A porosity gradient GDL with increased local porosity on the CL side boosts current density. Although compression reduces mean current density, at high compression ratios, it renders cathode performance akin to that of uniform GDLs.

The study concludes that the factors most influential to mean current density, ranked in order of impact, are overpotential, porosity, compression, porosity gradient, and pressure difference, with overpotential exerting the most substantial effect.

## Figures and Tables

**Figure 1 nanomaterials-14-02012-f001:**
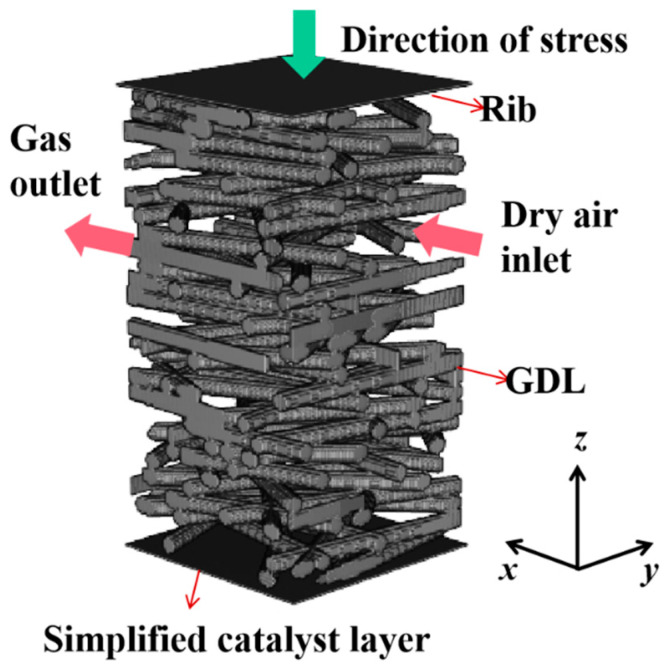
The schematic of the computational domain.

**Figure 2 nanomaterials-14-02012-f002:**
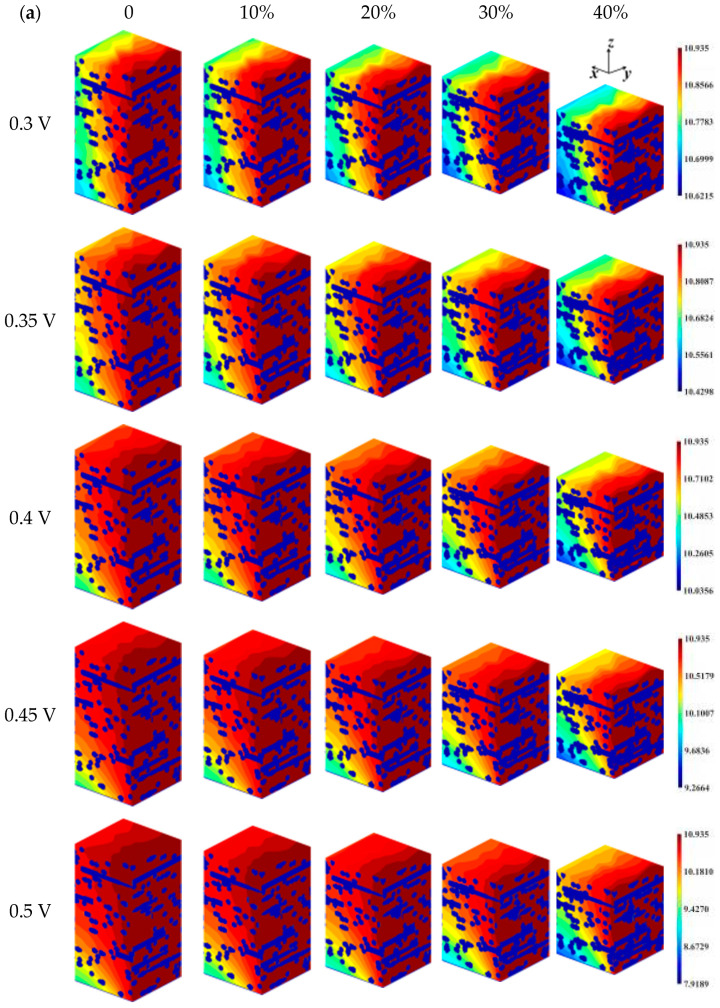
Concentration distribution within the GDL for different overpotentials and compression ratios: (**a**) oxygen; (**b**) water vapor.

**Figure 3 nanomaterials-14-02012-f003:**
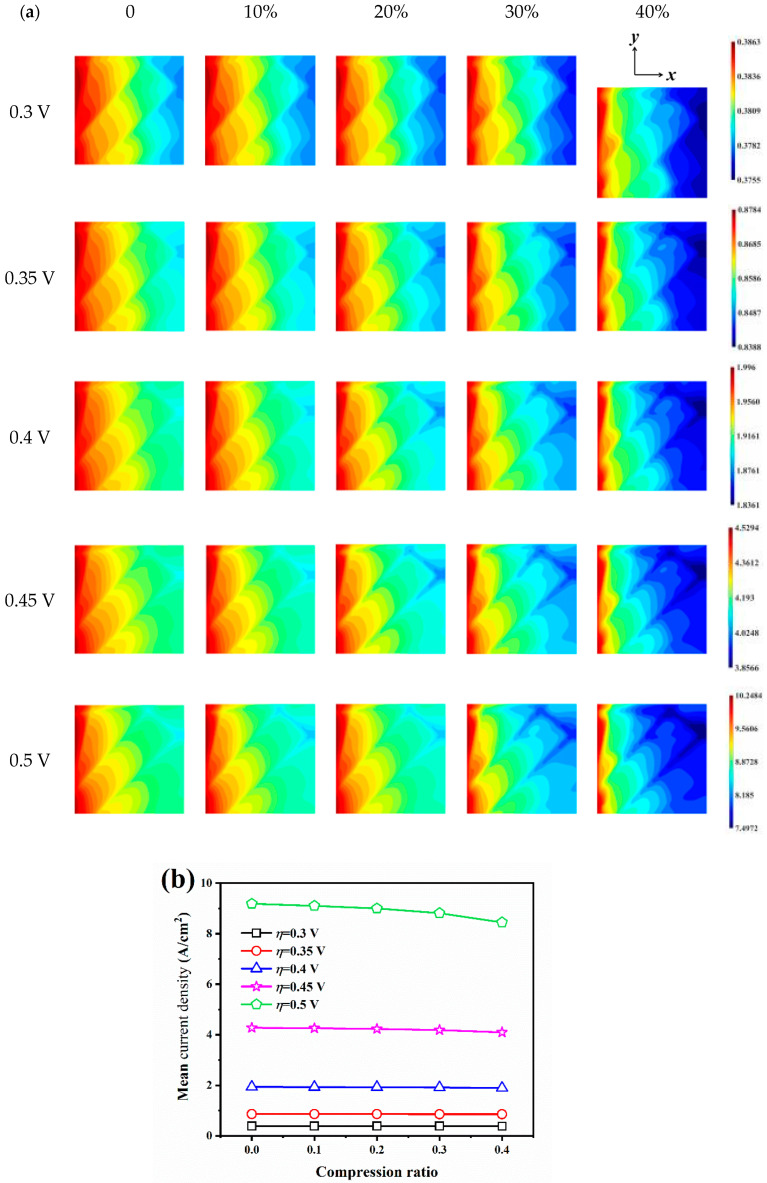
The current density on the CL surface for different overpotentials and compression ratios: (**a**) current density distribution; (**b**) mean density distribution.

**Figure 4 nanomaterials-14-02012-f004:**
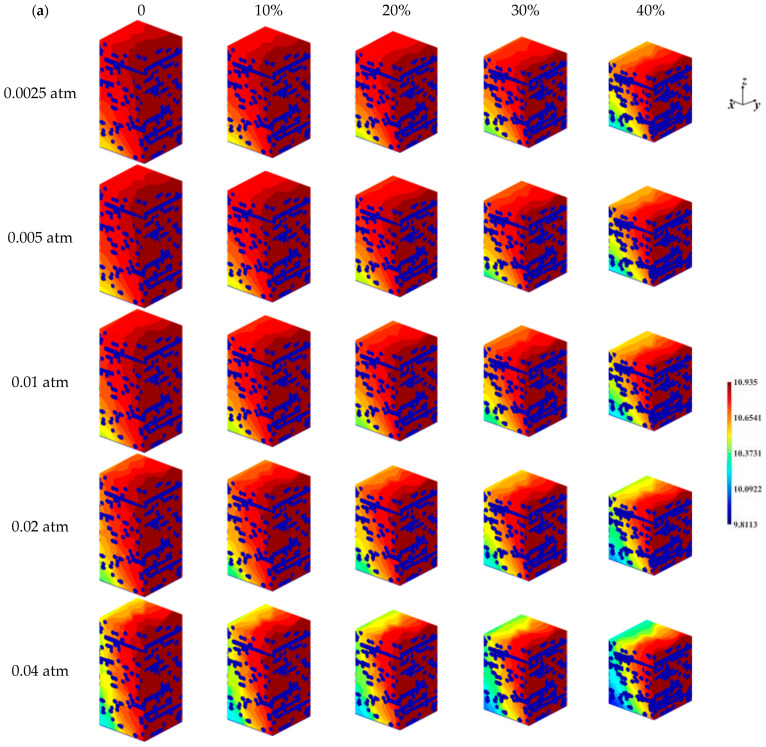
Concentration distribution within the GDL for different pressure differences and compression ratios: (**a**) oxygen; (**b**) water vapor.

**Figure 5 nanomaterials-14-02012-f005:**
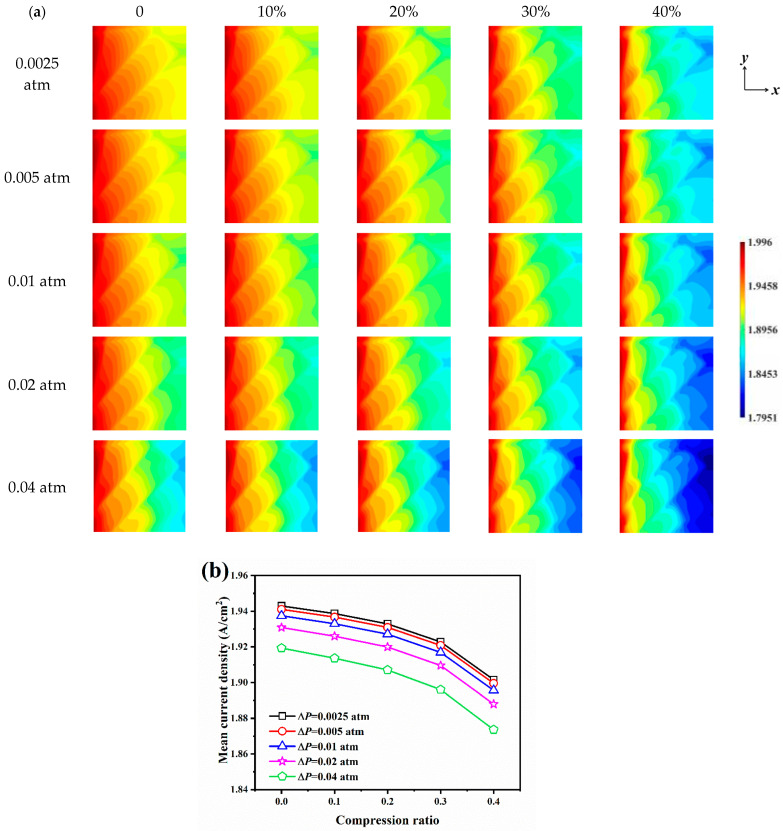
The current density on the CL surface for different pressure differences and compression ratios: (**a**) current density distribution; (**b**) mean density distribution.

**Figure 6 nanomaterials-14-02012-f006:**
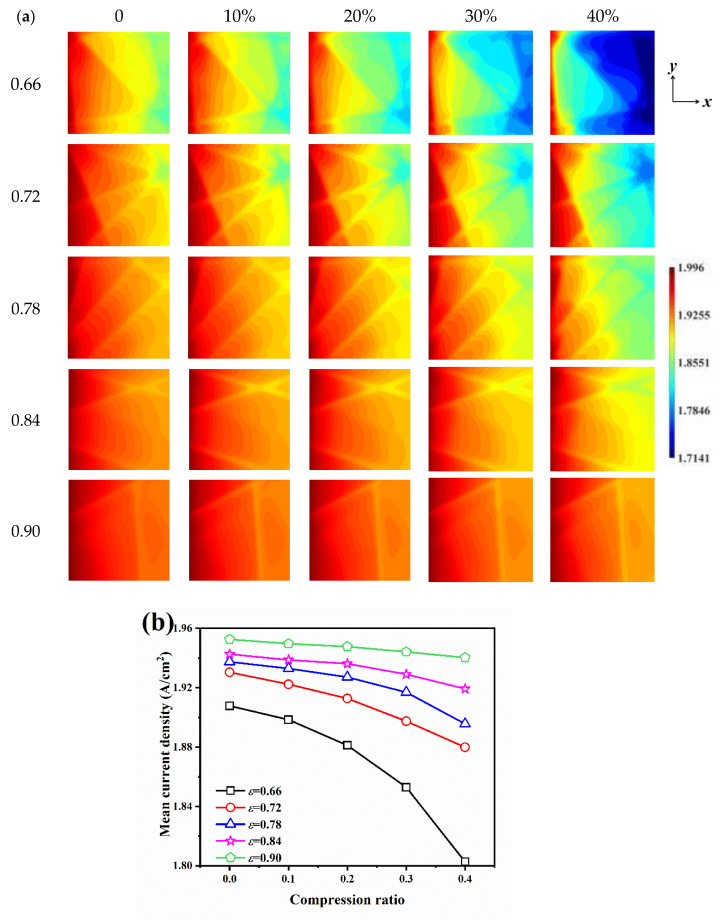
The current density on the CL surface for different porosities and compression ratios: (**a**) current density distribution; (**b**) mean density distribution.

**Figure 7 nanomaterials-14-02012-f007:**
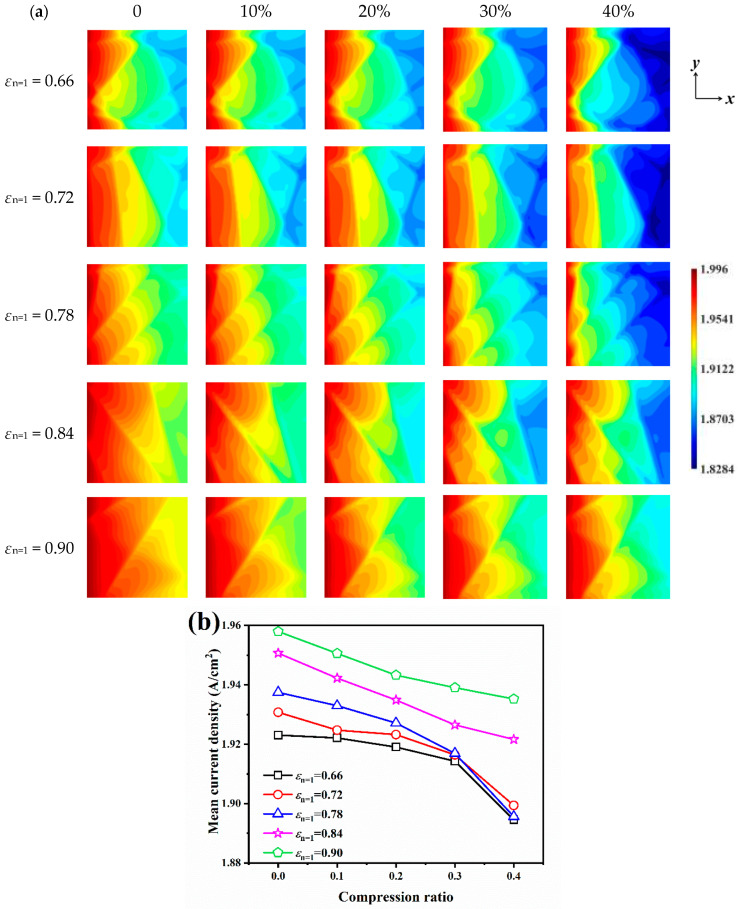
The current density on the CL surface for different porosity gradients and compression ratios: (**a**) current density distribution; (**b**) mean density distribution.

**Figure 8 nanomaterials-14-02012-f008:**
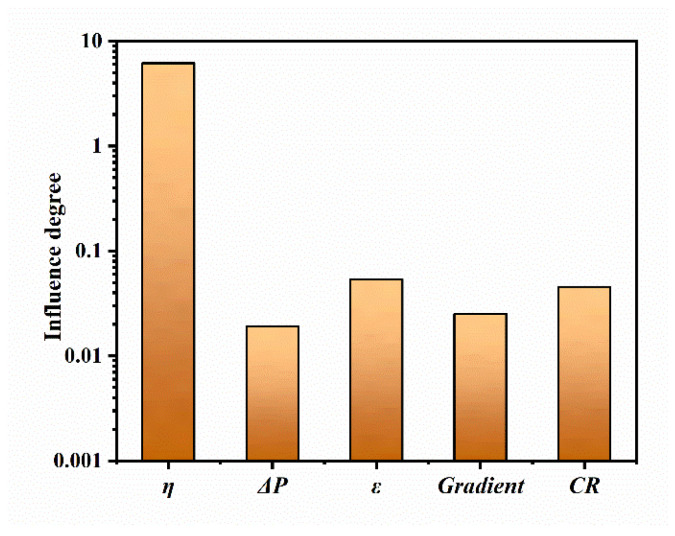
Degree of influence of different parameters on mean current density.

**Table 1 nanomaterials-14-02012-t001:** Base and calculated conditions for the considered parameters.

Parameters	Base Case	Calculated Cases
Overpotential (*η*)	0.4 V	0.3, 0.35, 0.45, 0.5 V
Pressure difference (Δ*P*)	0.01 atm	0.0025, 0.005, 0.02, 0.04 atm
Porosity (*ε*)	0.78	0.66, 0.72, 0.84, 0.90
Porosity gradient	uniform	Step type: *ε*_n=1_ = 0.66, 0.72, 0.84, 0.90

## Data Availability

Data are contained within the article.

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
