# Peer review of "Impact of Gas Diffusion Layer Compression on Electrochemical Performance in Proton Exchange Membrane Fuel Cells: A Three-Dimensional Lattice Boltzmann Pore-Scale Analysis"

_nanomaterials, 2024, doi:10.3390/nano14242012_

Round 1

Reviewer 1 Report

Comments and Suggestions for Authors

A three-dimensional lattice Boltzmann model is used to examine how gas diffusion layer (GDL) compression affects PEMFC electrochemical performance. Compression changes pore structure, affecting oxygen and water vapor transport and cathode current density. Overpotential and porosity are the main factors that reduce oxygen accessibility, water vapor retention, and mean current density when compression is increased. These findings help optimize GDL design for PEMFC performance in real-world conditions. However, some minor modifications are required.

1.     Does the study thoroughly examine how compression affects the concentration of oxygen, distribution of water vapor, and current density in GDL?

2.     Is the impact of compression ratio on electrochemical performance sufficiently addressed, especially under high overpotential conditions?

3.     Does the paper sufficiently address the practical implications of its findings for the design of PEMFCs in marine environments or other applications?

Author Response

Special thanks to the reviewers and editors for their contributions to this manuscript. I revised the manuscript as soon as I received the review comments, and my response to the reviewers is as follows:

Comment 1. Does the study thoroughly examine how compression affects the concentration of oxygen, distribution of water vapor, and current density in GDL?

Response: The presence of a denser microstructural partition in the GDL when compressed prevents the diffusion of water vapour from the GDL and the diffusion of oxygen to the CL, which is responsible for the attenuation of the current density. The above factors were clearly stated in the manuscript.

Comment 2. Is the impact of compression ratio on electrochemical performance sufficiently addressed, especially under high overpotential conditions?

Response: The increase in mass transfer resistance is inevitable when the GDL is compressed, and one of the effective ways to solve the problem of current density decay at high compression ratios is to improve the mass transfer efficiency by rationally designing the GDL structure. As shown in section 3.4 of the manuscript, a reasonable porosity gradient can improve mass transfer and result in higher current density even at high compression ratios.

Comment 3. Does the paper sufficiently address the practical implications of its findings for the design of PEMFCs in marine environments or other applications?

Response: As described in the INTRODUCTION section of the manuscript, PEMFCs can power a wide variety of vehicles, including forklifts, cars, ships, motorcycles, submarines, etc., and compression is a common phenomenon that occurs in fuel cell stacks due to preloading of bolts during assembly and operation. This manuscript does not focus solely on the study of PEMFCs in a specific application scenario, but hopes to explore the common problem of GDL compression.

Reviewer 2 Report

Comments and Suggestions for Authors

This paper presents the results of the effects of overpotential, porosity, compression, porosity gradient, and pressure difference on the dynamics of the processes taking place in compressed gas diffusion layers of proton exchange membrane fuel cells. The model used is quite complete, considering all the points relevant to the low-temperature fuel cells operation, what would be very useful to optimize their performance. The model used leads to reasonable results, with the corresponding physicochemical characteristics well defined in the figures of the paper with colored pictures related to the properties gradients. The paper is well organized and understandable, separating the study of the different variables to see specific effects. For this reason, I find the paper suitable for publication as it is.

Author Response

We are very grateful to the reviewers for their positive comments on our manuscript, which is a spiritual motivation to support us to continue our in-depth research, and we wish the reviewers good luck and happiness in their work.